# Temporal and spatial relationship between gluteal muscle Surface EMG activity and the vertical component of the ground reaction force during walking

Christoph Anders[1]*, Klaus Sander[2], Frank Layher[2], Steffen Patenge[2¤], Raimund W. Kinne[3]

**1** Division of Motor Research, Pathophysiology and Biomechanics; Experimental Trauma Surgery; Department for Hand, Reconstructive, and Trauma Surgery; Jena University Hospital; Friedrich Schiller University Jena, Jena, Germany, **2** Chair of Orthopedics, Department of Orthopedics, Jena University Hospital, Waldkliniken Eisenberg GmbH, Deutsches Zentrum für Orthopädie; Friedrich Schiller University Jena, Jena, Germany, **3** Experimental Rheumatology Unit, Department of Orthopedics, Jena University Hospital, Waldkliniken Eisenberg GmbH, Deutsches Zentrum für Orthopädie; Friedrich Schiller University Jena, Jena, Germany

¤ Current address: General Practice Dr. Baumbach, Apolda, Germany
* christoph.anders@med.uni-jena.de

**Data Availability Statement:** All relevant data are within the manuscript and its Supporting Information files.

## Abstract

### Background

Optimized temporal and spatial activation of the gluteal intermuscular functional unit is essential for steady gait and minimized joint loading.

### Research question

To analyze the temporal relationship between spatially resolved surface EMG (SEMG) of the gluteal region and the corresponding ground reaction force (GRF).

### Methods

Healthy adults (29♀; 25♂; age 62.6±7.0 years) walked at their self-selected **slow**, **normal**, and **fast** walking speeds on a 10 m walkway (ten trials/speed). Bilateral paired eight-electrode strips were horizontally aligned at mid-distance of the vertical line between greater trochanter and iliac crest. Concerning the ventral to dorsal direction, the center of each strip was placed on this vertical line. Initially, these signals were monopolarly sampled, but eight vertically oriented bipolar channels covering the whole gluteal region from ventral to dorsal (P1 to P8) were subsequently calculated by subtracting the signals of the corresponding electrodes of each electrode strip for both sides of the body.

Three vertical bipolar channels represented the tensor fasciae latae (TFL; P2), gluteus medius (Gmed, SENIAM position; average of P4 and P5), and gluteus maximus muscles (Gmax; P7).

**Funding:** RWK: FKZ 035577D, 0316205B, and 13N12601 Bundesministerium für Bildung und Forschung www.bmbf.de The funders had no role in study design, data collection and analysis, decision to publish, or preparation of the manuscript.

To determine the interval between SEMG and corresponding GRF, the time delay (TD) between the respective first amplitude peaks (F1) in SEMG and vertical GRF curves was calculated.

## Results

Throughout the grand averaged SEMG curves, the absolute amplitudes significantly differed among the three walking speeds at all electrode positions, with the amplitude of the F1 peak significantly increasing with increasing speed. In addition, when normalized to **slow**, the relative SEMG amplitude differences at the individual electrode positions showed an impressively homogeneous pattern.

In both vertical GRF and all electrode SEMGs, the F1 peak occurred significantly earlier with increasing speed. Also, the TD between SEMG and vertical GRF F1 peaks significantly decreased with increasing speed.

Concerning spatial activation, the TD between the respective F1 peaks in the SEMG and vertical GRF was significantly shorter for the ventral TFL position than the dorsal Gmed and Gmax positions, showing that the SEMG F1 peak during this initial phase of the gait cycle occurred earlier in the dorsal positions, and thus implying that the occurrence of the SEMG F1 peak proceeded from dorsal to ventral.

## Significance

Tightly regulated spatial and temporal activation of the gluteal intermuscular functional unit, which includes both speed- and position-dependent mechanisms, seems to be an essential requirement for a functionally optimized, steady gait.

## Introduction

During the stance phase of locomotion, the bilateral gluteal muscle units mainly consisting of tensor fasciae latae (TFL), gluteus medius (Gmed), and gluteus maximus (Gmax), alternatingly perform the following functions: i) stiffening of the hip joint; ii) pulling of the body over the heel rocker; iii) stabilization of the ipsilateral hip during the single-support stance phase against subsidence of the contralateral hip; and, finally, iv) initiation and execution of leg propulsion and subsequent swing phase [1,2]. This repetitive function requires a finely tuned and coordinated activity of these gluteal muscles. As a consequence, temporally and spatially differentiated and changing functional characteristics of gluteal muscles are needed during locomotion, resulting in a heterogeneous Surface EMG (SEMG) signal across the gluteal intermuscular functional unit [3,4].

In the early stance phase during undisturbed walking, the dorsal parts of the gluteal muscle unit (but possibly also ventral muscles such as the TFL) and lower extremity muscles [2] contribute to stiffening and/or extension of the hip joint during the loading response, which results in the well-known ground reaction forces [2,5–7]. During the single-legged mid-stance, the medial parts of the gluteal unit prevent the contralateral hip from subsidence [1], whereas the preparation and execution of the swing phase relies on the anterior parts of the gluteal muscle unit (e.g. the TFL; [1]), together with other, deeper hip flexor muscles such as the psoas major and iliacus.

The individual components of leg locomotion can only be fully evaluated by analyzing the GRF in the three directions horizontal (x), lateral (y), and vertical (z; [5,6]). The vertical GRF (z) is characterized by a triphasic pattern representing the weight acceptance peak during load response and initial single support phase (F1), followed by a mid-stance valley (F2) during forward movement of the opposite leg, and a second terminal stance peak (F3) during movement of the body weight over the forefoot rocker [2]. This basic pattern is the result of the acting forces of all involved muscles, which are also reflected in the SEMG activation profile of the gluteal muscles, however with spatial alterations across the gluteal region [3,7]. In terms of clinical relevance, GRF analyses have been used to validate numerous gait abnormalities, e.g. osteoarthritis, including the specific analysis of acting forces and affected joints by applying inverse dynamics approaches [8].

When measuring in vivo muscle activity by SEMG, however, it has to be kept in mind that the mechanical result of any measured EMG signal is delayed to a certain extent–resulting in the electromechanical delay (EMD; [9]). The EMD shows a range of values between 30 and 100 ms for different muscles [10–13], and an EMD of about 50–66 ms can be assumed from the only two publications reporting on the Gmed [12,14]. Since direct definition of the EMD, i.e. the determination of the lag between a superthreshold EMG activity and the respective force signal in specific muscles, was not available in the present study, the time delay (TD) between the first peak in SEMG and the well established F1 peak of the GRF curves was used.

By simultaneous temporal analysis of the spatially resolved SEMG of the gluteal region and the corresponding GRF, the present study pursued the question which adjustments are required to achieve fine-tuned temporal and spatial activation of the gluteal muscles for optimized gait and joint loading. The present study design with an older-aged healthy control cohort was deliberately chosen to match the mean age of patients developing hip osteoarthritis and therefore with a higher risk to undergo total hip arthroplasty. Indeed, this study demonstrated regionally differentiated and strongly regulated muscular activation patterns representing the different components of the gluteal intermuscular functional unit and their respective sub-regions during the changing demands of the gait cycle. These findings provide a substantial basis for further extended studies in locomotion physiology and pathophysiology.

Potential clinical implications of these novel results include: i) improved basis for detailed (pre-operative) diagnosis of hip joint locomotion predisposing the individual to develop osteoarthritis [15,16]; ii) close and multi-component monitoring of the post-operative improvement of hip function after endoprosthetic replacement surgery [17,18]; iii) bio-feedback optimization of post-operative rehabilitation by considering the close relationship between biomechanical demands and finely tuned activation of the acting muscles [19], including the timing, loading intensity, and walking speed of the applied procedures, as well as the usage of treadmills [3,20]; v) definition of differentiated training designs for neurological dysfunctions or coordination training for the elderly [21].

## Methods

Healthy and age-matched older adults of both genders were investigated (29 females and 25 males; age range from 50 to 75 years; for details see Table 1). Participants were recruited from University Hospital employees and their acquaintances. Height, weight, and BMI of the age-matched females and males in the present study were significantly different (Table 1, [3,20]).

All participants were informed about study purpose and procedure, and written informed consent was obtained from every volunteer. The study was approved by the local ethics committee (3002-12/10). To exclude relevant orthopedic and neurologic disorders, participants were clinically investigated and further queried about their medical history. Besides general

**Table 1. Participant characteristics.**

|  | Female (n = 29) | Male (n = 25) | Statistics |
|---|---|---|---|
| Age [years] | 63.7 ±6.8 | 61.3 ±7.0 | n. s. |
| Height [cm] | 164.0 ±5.9 | 178.3 ±6.5 | p < 0.001 |
| Weight [kg] | 68.8 ±11.5 | 90.2 ±12.1 | p < 0.001 |
| BMI [kg/m$^2$] | 25.5 ±4.4 | 28.3 ±3.5 | p = 0.010 |

BMI: Body mass index; data are given as means ± SD; n. s.: Not significantly different.

health problems possibly interfering with unrestricted study participation, specific orthopedic exclusion criteria were hip or knee endoprosthetic joint replacements, pain during locomotion, or clinical signs of knee or hip osteoarthritis.

Part of the SEMG data of this group was already published in previous studies [3,20].

## Locomotion investigation

After SEMG instrumentation (see below), participants walked at their self-selected **slow** (3.2 ± 0.6 km/h; mean ± SD; Min: 1.7, Max: 4.4 km/h), **normal** (5.2 ± 0.6 km/h; Min: 3.8, Max:. 6.6 km/h), and **fast** walking speeds (6.2 ± 0.8 km/h; Min: 4.8, Max: 8.1 km/h [20]) on a 10 m walkway (at least ten trials per walking speed resulting in a minimum of 50 complete strides per walking speed).

These walking speeds, calculated by dividing the travelled distance derived from an optical marker-based motion tracking system (Vicon 460, Oxford Metrics Ltd., Oxford, GB) by the time difference between two subsequent heel contacts derived from the force plates (threshold 20 Newton in GRFz direction; as pre-determined in the Vicon 460 system; personal communication, Oxford Metrics Ltd., Oxford, GB), are in good agreement with those in published data [3,22]. The normal walking speed was chosen as the first speed, subsequently followed by the slow and fast walking speeds. In each trial, steady walking conditions were visually identified and marked in the EMG file by one experienced examiner (S.P.). Respective start (end of the acceleration phase) and stop markers (beginning of the deceleration phase) were manually placed in the EMG data to define the time window for the steady walking condition and the inclusion of the respective heel strikes for subsequent analyses.

For GRF assessment, the walkway was equipped with three embedded, centrally placed force plates (once: Kistler–Kistler Instruments AG, CH; twice: AM-TI–AMTI, US). While walking over these force plates, the GRF in the x, y, and z directions was determined. For the current investigation, we focused on the vertically directed component of the GRF (z) for the following reasons: i) this GRF direction represents the quantitatively most prominent part of the GRF; ii) it most closely represents the main function of the gluteal muscles, i.e. providing the horizontal alignment of pelvis and trunk during the stance phase; iii) it shows the highest similarity with the SEMG profiles of the gluteal muscles (see below); and iv) it is the only GRF direction which showed a substantial temporal change (i.e., > 5% points) for its F1 peak when comparing the normal or fast to the slow walking speed (Table 2, S1 Fig).

To ensure a normal, undisturbed gait pattern, the participants were not informed about existence, position or function of the force plates. Targeting of complete strides on the centrally located force plates was achieved by individually adjusting the starting point according to the respective individual gait pattern of every participant.

For SEMG data collection, two electrode strips, each containing eight surface electrodes (H938SG, Covidien; Ag-AgCl, circular uptake area of 2 cm$^2$; inter-electrode distance of 2.5 cm; Fig 1) were horizontally aligned at mid-distance of the vertical line between the greater

**Table 2. Amplitude and time characteristics (in the latter case expressed as either % gait cycle or as % point difference vs. the slow walking speed) of the F1 components for all GRF directions at slow, normal, and fast walking speeds.**

| Walking speed | Parameter | GRF x | GRF y | GRF z |
|---|---|---|---|---|
| Slow | Relative amplitude [% body weight] | 12.2 ± 3.1 | 3.2 ± 1.5 | 100.0 ± 4.7 |
| | Time [% gait cycle] | 11.7 ± 2.2 | 4.2 ± 1.0 | 19.9 ± 2.6 |
| Normal | Relative amplitude [% body weight] | 20.7 ± 4.1[§] | 5.4 ± 2.2[§] | 118.5 ± 9.7[§] |
| | Time [% gait cycle] (% point diff. vs. Slow) | 10.2 ± 1.4[§] (-1.4 ± 2.3) | 4.4 ± 1.0 (0.2 ±0.8) | 14.1 ± 1.4[§] (-3.8 ± 2.5) |
| Fast | Relative amplitude [% body weight] | 25.2 ± 4.8[§ $] | 6.4 ± 2.9[§ $] | 132.0 ± 12.1[§ $] |
| | Time [% gait cycle] (% point diff. vs. Slow) | 9.6 ± 1.7[§ $] (-2.0 ± 2.6) | 4.0 ± 1.0[§] (-0.2 ± 0.9) | 12.8 ± 2.2[§ $] (-5.1 ± 3.1) |

§: $p < 0.01$ vs. Slow

$: $p < 0.01$ vs. Normal (including Bonferroni correction).

Data are given as means ± SD. The time differences vs. Slow were calculated on a subject-individual basis as percentage point differences.

a)

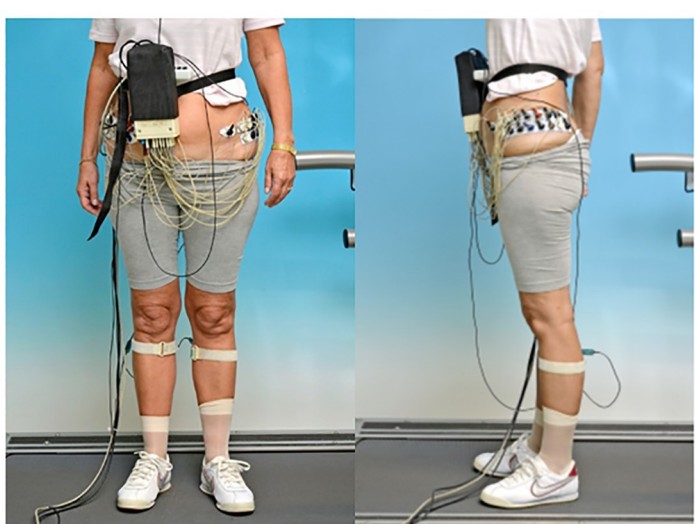

b)

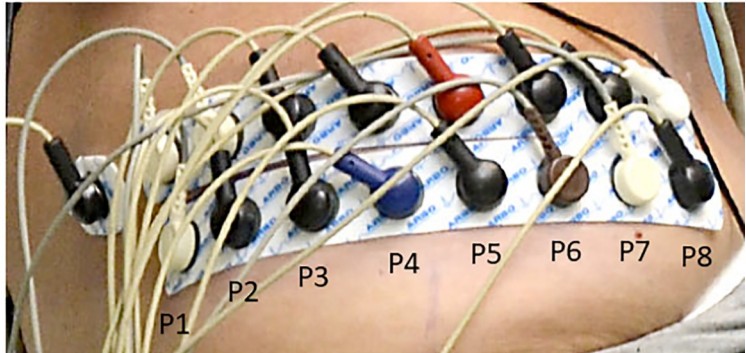

**Fig 1. Applied electrode strips together with the respective cables and amplifier.** A) frontal and side view of a participant, b) enlarged view of the electrode strips and the electrode position labels. The single electrode is one of the two conjoined reference electrodes above the anterior superior iliac spine position on both sides.

trochanter and the iliac crest on both sides of the body [3]. Concerning the ventral to dorsal direction, the center of each strip was placed on the vertical line between the greater trochanter and the iliac crest. The electrodes (named P1 to P8 from the ventral to the dorsal margin) covered the whole gluteal region [3], including TFL (electrode position P2 [23]), Gmed (SENIAM position; average of P4 and P5; [24]), and Gmax (P7 [25]).

Signals were monopolarly captured (using bilateral conjoined electrodes at the anterior superior iliac spine as reference), amplified (gain 1000, -3 dB at 5 Hz and 700 Hz, DeMeTec), analog-to-digital converted (Tower of Measurement—ToM; 2048 samples/s, DeMeTec), and stored on hard disk for off-line analysis (ATISArec, GJB). In addition, pressure sensors (FSR-402, Interlink electronics) were placed directly below each heel to allow reliable heel contact identification. The signal of the pressure sensors was simultaneously measured and stored together with the SEMG data set in one file (i.e. these data were synchronized).

## SEMG data analysis

First all heel contact events were separately identified in the SEMG data set using custom MATLAB-scripts.

Bipolar channels were calculated from the monopolar signals by subtracting the signals from the corresponding electrode numbers of each electrode strip for both sides of the body. SEMG data were band-pass filtered between 20 Hz and 400 Hz. To account for the possibility of randomly occurring interferences from the electrical current supply, a 50 Hz (band width ± 1 Hz) notch filter was applied.

SEMG data were quantified as root mean square (RMS) values that were smoothed with an overlapping, sample wise moving rectangular window of 50 ms (overlap 49.5 ms), and time normalized (100% = one complete stride) on the basis of two subsequent ipsilateral heel strikes (time resolution: 0.5%; 201 time points). To provide a reliable data basis for the calculation of grand averaged amplitude curves for each participant and each speed, only strides that did not exceed 10% deviation from the individually determined median stride time were considered. Also, outliers due to artefacts resulting in irregular SEMG curve patterns (i.e., exceeding the limits of two standard deviations from the mean) were identified by visual inspection and deleted. RMS amplitude levels in grand averaged curves were expressed either as mean absolute values for all participants or as relative differences following normalization to the slow walking speed.

As the centers of the eight-electrode strips were located perpendicularly above the greater trochanter, a virtual bipolar channel was calculated by averaging the RMS values of the 4[th] and 5[th] bipolar channels, that matched the SENIAM-recommended electrode position for the Gmed muscle [26].

Although the F1, F2, and F3 peaks and valleys in the z direction of the GRF were easily identifiable, the corresponding characteristics of the SEMG curves were influenced by walking speed and electrode position and thus more difficult to pinpoint. Thus the analyses for the TD, i.e., the delay between the SEMG and GRF curves, were only performed for the F1 peak of the GRF curves and the first peak of the SEMG (i.e., the maximum value in the time window between heel strike and 25% of the normalized gait cycle). As for the GRF data, this was done separately for every participant, walking speed, and electrode position using the grand averaged amplitude curves.

Amplitude and time characteristics of the F1 peak of the SEMG were also calculated on a subject-individual basis. Amplitudes were then expressed as absolute values or as relative values following normalization to the slow walking speed. The time point of the occurrence of the F1 peak was expressed as a percent value of the total stride time for the different walking

speeds (1.33 ± 0.18 s for slow, 1.00 ± 0.08 s for normal, and 0.92 ± 0.08 s for fast). TD data (reflected in the delay between the first peak in the SEMG curve and GRF F1 peaks) was expressed either as absolute delay (ms) or as relative delay (% of stride time).

### Statistical analyses

For the amplitude curves and the amplitude differences following normalization to the slow walking speed, paired t-tests were performed to assess statistical effects of the different walking speeds throughout the gait cycle (p<0.05). For the present testing of dependent groups, a sample size of 34 would have been sufficient for a medium effect size of 0.5 (two-tailed probability, power 0.8; calculation using G*Power, V 3.1.9.4). To avoid the accumulation of a type I statistical error due to multiple testing (in particular caused by the high number of time points), i.e., to falsely reject the respective null hypotheses, these test results were corrected for the effects of multiple testing using the stepwise Bonferroni-Holm procedure [3,27].

Further statistical analyses dealt with the occurrence of and the time difference between the F1 peaks in SEMG and GRF and consisted of a repeated measures ANOVA with side (left, right; 2), walking speed (slow, normal, fast; 3), and position (P1 toP8; 8) as the within subject parameters and gender as the between subject parameter. To address intra-individual amplitude differences, relative amplitude changes of the F1 peak after normalization to the slow walking speed were also analyzed using paired t-tests.

Neither body side nor gender had any significant and relevant influence on the amplitude and time characteristics of the F1 peak in the SEMG and GRF curves (effect sizes -partial Eta$^2$- all below 0.14, i.e., the threshold for large effects [28]). Thus, the results show representative data for the left side, pooled for all participants independently of their gender.

In contrast, the main effects of walking speed and electrode position showed significant and relevant interactions for the amplitude and time characteristics of the F1 peak in SEMG and GRF (all p values < 0.001; all effect sizes > 0.18).

### Results

The GRF in the vertical (z) direction contained the typical early F1 peak at the transition from the weight acceptance to the single support phase, the mid-stance F2 force valley, and the second F3 peak during terminal stance, with the respective significant differences among speeds for the F1 and F2 peak (Fig 2A). This basic tri-phasic pattern was also recognizable in the SEMG activation profiles for the TFL (P2), Gmed (P4/5), and Gmax (P7), however with decreasing similarity of the F2 and F3 components from ventral to dorsal (Fig 2A). Throughout the grand averaged SEMG curves, the absolute amplitudes significantly differed among the three walking speeds at all electrode positions. Similar to the GRF, the amplitude of the F1 peak significantly increased with increasing speed (Fig 2A; also compare with Fig 3). Again in parallel with the GRF, the SEMG F1 peak occurred earlier with increasing speed (Fig 2A). When normalizing the amplitudes in the grand averaged SEMG curves to those at the **slow** walking speed, the resulting relative amplitude differences for the individual electrode positions showed an impressively homogeneous alteration pattern, suggesting a tightly regulated, speed-dependent spatial activation of the gluteal intermuscular functional unit (Fig 2B).

Upon detailed analysis of both GRF and SEMG on a subject-individual basis, the significant increase of the F1 peak with increasing speed was fully confirmed (Fig 3A and 3B). In addition, the F1 peak in GRF and SEMG for all electrode positions occurred significantly earlier with increasing speed (Fig 3C).

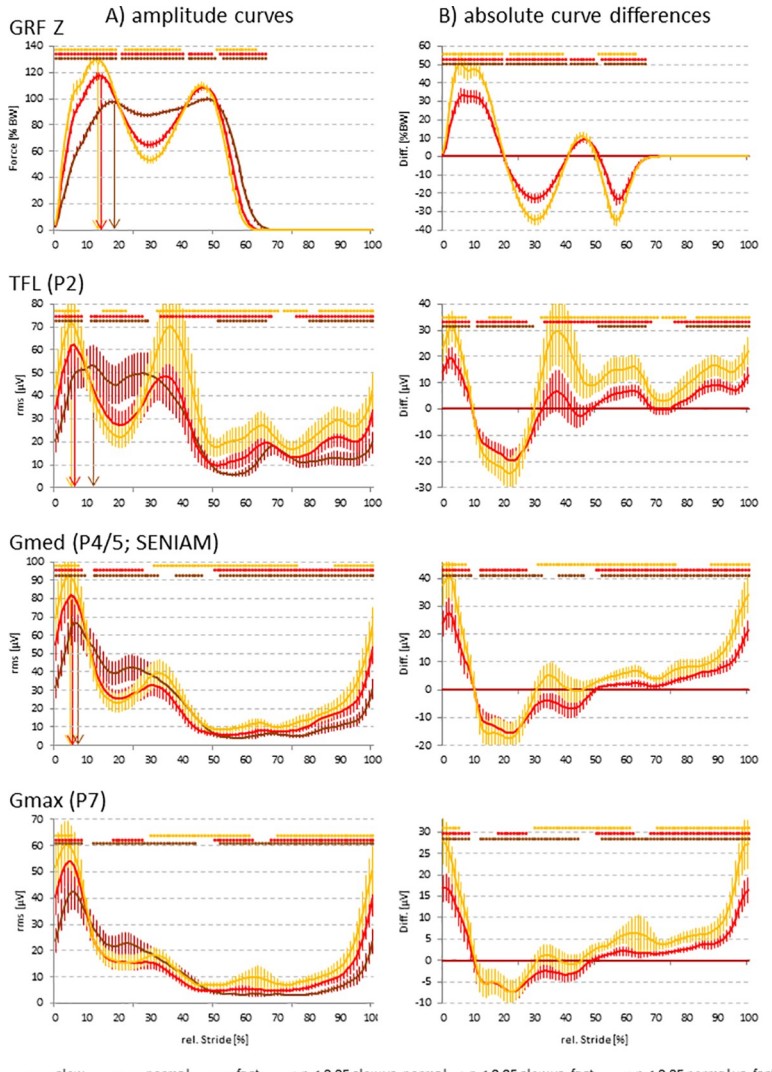

**Fig 2.** Grand averaged curves (A) and curve differences following normalization to the slow walking speed (B) for GRF z and SEMG for the TFL (P2), Gmed (P4/5; SENIAM), and Gmax (P7) positions. Significant differences among the individual walking speeds are indicated by colored bars (p < 0.05; Bonferroni-Holm procedure). The arrows reflect the temporal occurrence of the F1 components during the gait cycle. Data are given as mean values ± 95% confidence intervals.

Also, the TD (reflected in the delay between the SEMG and GRF F1 peaks) significantly decreased with increasing speed (Fig 4), which was more pronounced when displaying the data as time differences in ms (Fig 4B).

Concerning spatial characteristics, the TD was significantly smaller for the ventral TFL position (P2) than the dorsal Gmed and Gmax positions (P4/5; P7; Fig 4A and 4B). In the case of the slow walking speed, significant differences were observed among all three electrode positions (Fig 4A and 4B).

## Discussion

A striking similarity was observed between the curve patterns in the vertical GRF and SEMG (most pronounced for the ventral TFL position P2), suggesting a tight physiological coupling

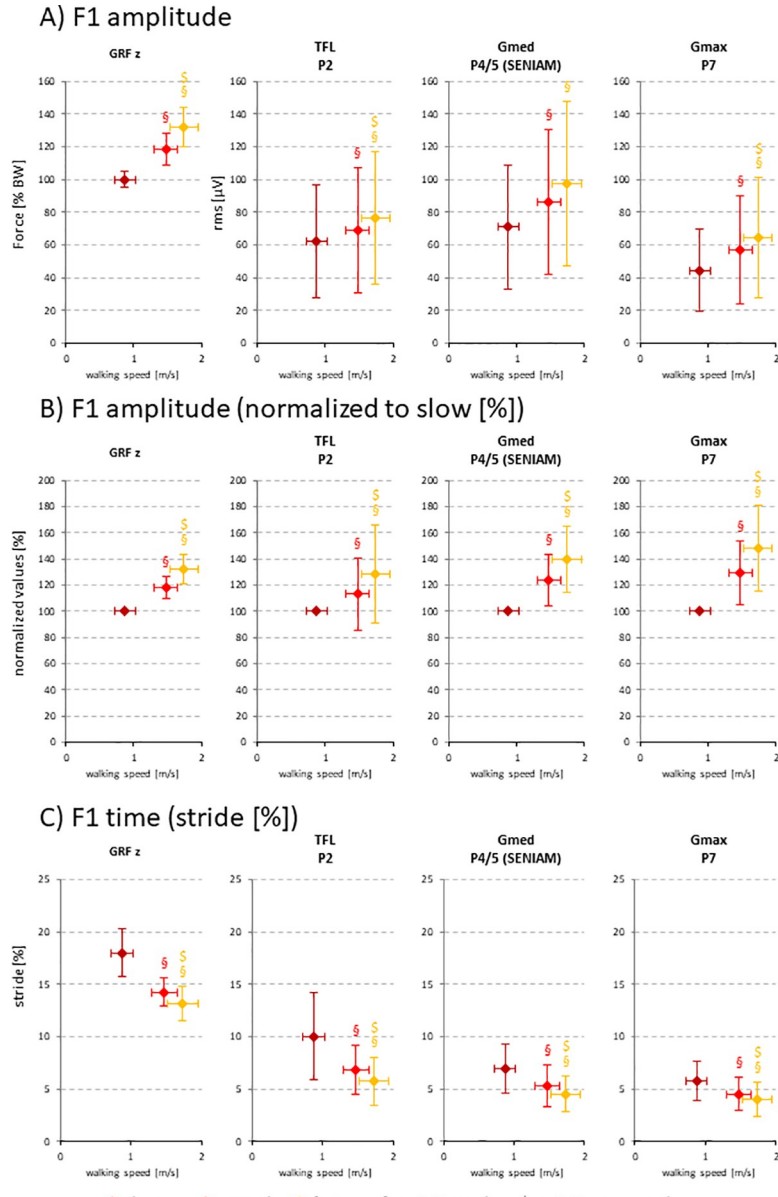

**Fig 3.** Mean F1 peak amplitudes (A), mean F1 peak amplitudes normalized to slow walking speed (B), and occurrence of F1 peaks during the normalized stride (C). Data are calculated on a subject-individual basis and given as mean values ± SD. Significant differences among the three walking speeds are indicated: § vs. slow, $ vs. normal. All p values are < 0.05 (including Bonferroni correction).

between the activation of gluteal muscles and their function in the compensation of the acting ground reaction forces and the stabilization of the hip joint [7].

As in the curves for the GRF, the F1 peak of the SEMG curves for all electrode positions significantly increased and occurred significantly earlier with increasing speed [3]. This has been observed previously, however without detailed analysis of the temporal occurrence of the SEMG F1 peak and the time delay between the F1 peak in SEMG and GRF [29]. Indeed, the F1 peaks in the SEMG always occurred substantially earlier than the respective F1 peak in the GRF, at least partially as a result of the physiological EMD [9]. This time gap became significantly smaller with increasing walking speed, indicating a clear influence of the speed. Also,

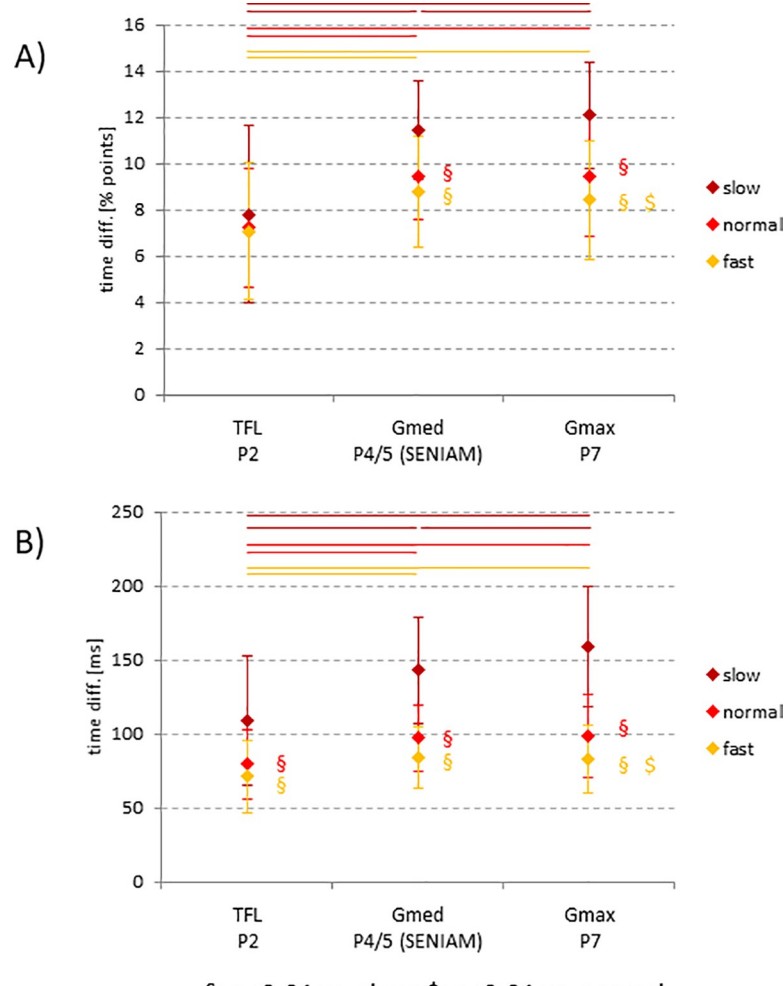

**Fig 4. Time differences between SEMG F1 peaks and the respective F1 peak in the GRF z curve (calculation on a subject-individual basis; provided as mean values ± SD).** A) time difference in % points of the normalized stride; B) time difference in ms. Significant differences among the three walking speeds are indicated: § vs. slow, $ vs. normal. All p values are < 0.05 (including Bonferroni correction). Significant differences among the three muscles are indicated by colored bars (p < 0.05; including Bonferroni correction).

this gap was significantly smaller for the ventral TFL position than the dorsal Gmed and Gmax positions, implying that the occurrence of the SEMG F1 peak proceeded from dorsal to ventral. This has been previously reported for the time to peak concerning the dorsal, middle, and anterior subdivisions of the Gmed [4].

The time gap between the occurrence of the F1 peak in SEMG and GRF significantly decreased with increasing walking speed. This was possibly due to a combined influence of alterations in stride time, consciousness of gait, stochastic character of SEMG, and the transition from tonic to phasic activation behavior of the muscles with increasing speed [30,31]. Indeed, the absolute values for this time gap for the normal and fast walking speeds at the Gmed position P4/5 (i.e., 97 and 84 ms, respectively) were close to the values previously reported for the Gmed (50–66 ms; [12,14], indicating that the time gap approached the physiological limits of the EMD, which cannot be influenced consciously [9]. In contrast, the value for the slow speed at this position was approx. 150 ms, which indicates an influence of additional factors, in particular the more tonic muscle activity at the slow walking speed [30,31].

Despite these considerations, the time gap between the occurrence of the F1 peak in the SEMG and the GRF still became shorter when comparing the fast to the normal walking speed. This regarded both the relative time difference in percent points (i.e., those normalized to the stride time), but also the absolute time difference in ms (compare with Fig 3). This is a puzzling finding, since, assuming an invariable EMD and a progressive shortening of the absolute stride time with increasing walking speed [6,32], the relative time differences should become larger with increasing walking speed. One potential explanation, i.e., a more synchronized activity of the respective motor units at faster walking, resulting in a steeper slope of the SEMG signal on the way to the F1 peak and thus a shorter "time to peak" [4,33], was excluded by the analysis of the slopes of the respective SEMG curves (data not shown).

Assuming that the physiological limits of the EMD were reached at fast walking speed, the neuromuscular system at higher locomotion speeds only has the possibility to further anticipate the SEMG activation in order to guarantee the time necessary for electromechanical coupling. Whereas there is no published systematic investigation of Gmed activity addressing this question, own unpublished data indeed showed that the F1 peak of the SEMG curves for Gmed and Gmax occurred progressively earlier with increasing treadmill speeds, in the case of Gmax even reaching time points before the heel strike for walking at 8 km/h (S2 Fig). This agrees with the notion that the beginning of the stride is conveniently, but 'artificially' defined by the floor contact during walking and running, leading Jacquelin Perry to the statement about the nature of locomotion: "With one action flowing smoothly into the next, there is no specific starting or ending point" [2].

The TD between the respective F1 peaks in the SEMG and GRF curves was significantly shorter for the ventral TFL position (P2) than the dorsal Gmed and Gmax positions (P4/5; P7), showing that the SEMG F1 peak during this initial phase of the gait cycle occurred earlier in the dorsal positions, and thus suggesting that the activation of the gluteal compartment during the loading response showed a time kinetics proceeding from dorsal to ventral. This is in agreement with the biomechanical requirements to continuously stabilize and realign the hip joint during locomotion. During the early F1 peak of the GRF (loading response), an anteriorly directed torque acts on the hip joint, which has to be compensated by all parts of the gluteal muscle compartment, consistent with a clearly discernable, strong F1 SEMG peak at all analyzed electrode positions (compare with Fig 2; [2]). The slightly earlier occurrence of the F1 peak in the more dorsal Gmax and Gmed portions is consistent with the fact that Gmax [1] and, to some degree, also the central part of the Gmed [1,34], strongly contribute to the stiffening of the hip joint, a process of utmost importance during the early phase of the loading response [2]. The prominent role of the more dorsal Gmax and Gmed portions during this phase is also reflected in the finding that the position-dependent time shift of the F1 peak at all speeds considerably increased only from the P4 position (central Gmed) to the P1 position (TFL; S3 Fig).

The involvement of the TFL in the loading response has been described previously, with some specificity for subpopulations of subjects or two different anteromedial and posterolateral fiber compartments within the TFL [2,35]. In the current results for the TFL, rather based on detecting the activity of the posterolateral fibers [3], there was a clear F1 SEMG peak at all walking speeds, indicating a central role of these fibers for the loading response. This is in good agreement with the notion that these posterolateral fibers possess a better mechanical advantage for hip abduction and internal rotation and are active near the heel strike [35].

Of note, the latter considerations for the TFL are independent of effects occurring in temporal conjunction with the second F2 peak of the SEMG, when the anteriorly directed force vector has already changed to a posteriorly directed force vector. This time point is concurrent to the start of the extension of the hip joint [2,7], in agreement with the fact that the anterior

parts of the gluteal muscle unit (e.g. the TFL; [1]) are pivotal for the preparation and execution of the swing phase.

The current study provides normative data in a healthy older population, which may now be used for future investigations concerning age-related influences, improved functional diagnostics, and the monitoring of therapeutic and rehabilitative efforts. Simultaneous analysis of the spatially resolved and specifically timed activation of the gluteal intermuscular functional unit and the respective GRF may thus serve to optimize therapy, but may also help to prevent the progression of hip osteoarthritis by early detection of clinically compensated joint dysfunction.

## Limitations

In the present study considerable effort was spent to ensure steady walking conditions by: i) placing start and stop markers for inclusion of data in the analysis; ii) using only complete strides that did not exceed ± 10% of the respective median stride time; and iii) excluding strides containing irregular stride patterns exceeding the limits of two standard deviations from the mean. However, we cannot completely exclude remaining variations of the walking speed, i.e. phases of non-steady walking. Since all the above mentioned corrections were performed online during data collection and pre-analysis the deviation between the final results with and without the application of these corrections cannot be determined.

In the current study, the "gold standard" to determine the EMD, i.e., the selective detection of the physiological signal onset for the EMG and the respective force, could not be applied. This is due to the fact that the investigation of the highly complex and repetitive gait requires steady walking conditions after the initial acceleration phase and before the beginning of the deceleration phase. Indeed, the gluteal muscles remain active throughout different phases of the stride, due to their complex functional importance as abductors (all parts of the Gmed), external rotators (dorsal parts of Gmed and Gmax), internal rotators (TFL and anterior parts of Gmed), flexors (TFL and anterior parts of Gmed), and extensors (dorsal parts of Gmed and Gmax) of the limb [1,7]. It should also be mentioned that the EMD appears to be affected by age [36,37], qualifying the present results specifically for middle-aged to older people. In addition, some deeper gluteal muscles were not investigated (i.e. gluteus minimus, piriformis, and gemelli muscles), since SEMG can only reliably detect the activity of superficial muscles, limiting the applicability of the present data to the actually investigated muscles.

Heel contact events were separately determined for GRF and SEMG data (heel pressure sensors), since at the time of the investigation no EMG plugin was available for the gait analysis system. In addition, only one complete stride per trial can be detected with the force plates, while for the SEMG measurements the whole distance of steady gait was used to collect as many strides as possible due to the stochastic nature of the SEMG signal. Therefore, heel contact events may not be completely superimposable in GRF and SEMG. Assuming a slight, but systematic detection error between the two detection methods (maximally 1.0 to 0.5 ms for the slow to fast walking speeds; data not shown) and looking at the range of the TD between the respective F1 peaks in SEMG and GRF (between 160 to 70 ms), an error of max. 0.75% would arise, which in our eyes does not question the validity of the present results. Nevertheless, the comparability of the current absolute TD values with those of future studies may be limited.

The F1 to F3 components of the GRF z direction were used as well-established gait characteristics, which were clearly identifiable in both force plate data and SEMG of the gluteal muscles. Due to the stochastic nature of the SEMG signal, however, only the F1 peak was identified with sufficient certainty at all electrode positions to allow a reliable comparison with the respective F1 peak in the GRF. The relative contributions of additional components, for

example the functional limits of tendons and joints, have to our knowledge only been addressed in musculoskeletal model systems [7] and merit further attention in the future.

## Conclusions

Tightly regulated spatial and temporal activation of the gluteal intermuscular functional unit, which includes both speed- and position-dependent mechanisms, seems to be an essential requirement for a functional, steady gait resulting in optimized GRF. This is realized by an earlier occurrence of the F1 peak in both SEMG and GRF and a simultaneous decrease of the time gap (including the TD) between the F1 peaks in SEMG and GRF with increasing speed. In addition, the time gap between SEMG and GRF becomes significantly smaller from the ventral TFL position to the dorsal Gmed and Gmax positions, in accordance with the differential functional importance of these muscles for hip movement and stability during gait. All these changes contribute to a fine-tuned, concerted, spatio-temporal organization of the commonly innervated different gluteal muscles during gait, which is clearly functionally and physiologically coupled rather than anatomically organized [3,4]. In addition, the current study provides normative data in a healthy older population, which may now be used for future investigations concerning age-related influences, improved functional diagnostics, and the monitoring of therapeutic and rehabilitative efforts.

## Supporting information

**S1 Fig.** Grand averaged curves (A) and curve differences following normalization to the slow walking speed (B) for the GRF x, y, and z directions. Significant differences among the individual walking speeds are indicated by colored bars ($p < 0.05$; Bonferroni-Holm procedure). Data are given as mean values ± 95% confidence intervals.
(TIF)

**S2 Fig. Representative circular display of the time-normalized, grand averaged SEMG amplitude curves during treadmill walking at speeds between 4 km/h and 8 km/h (ipsilateral heel strike at 0% position; contralateral heel strike at 50% position; time progress clockwise).** Electrodes for Gmed and Gmax were positioned at the recommended SENIAM positions (http://www.seniam.org). Data are given as mean values.
(TIF)

**S3 Fig. Relative occurrence of the F1 peaks during the normalized stride for all SEMG positions.** Data are calculated on a subject-individual basis and given as mean values ± SD.
(TIF)

**S1 Data.**
(ZIP)

## Author Contributions

**Conceptualization:** Christoph Anders, Klaus Sander, Frank Layher, Raimund W. Kinne.

**Data curation:** Christoph Anders, Steffen Patenge.

**Formal analysis:** Christoph Anders, Klaus Sander, Frank Layher, Steffen Patenge.

**Investigation:** Christoph Anders, Klaus Sander, Frank Layher, Steffen Patenge.

**Methodology:** Christoph Anders, Klaus Sander, Raimund W. Kinne.

**Project administration:** Christoph Anders, Raimund W. Kinne.

**Resources:** Raimund W. Kinne.

**Software:** Christoph Anders.

**Supervision:** Christoph Anders, Raimund W. Kinne.

**Validation:** Christoph Anders, Klaus Sander, Frank Layher, Steffen Patenge.

**Visualization:** Christoph Anders.

**Writing – original draft:** Christoph Anders, Klaus Sander, Frank Layher, Steffen Patenge, Raimund W. Kinne.

**Writing – review & editing:** Christoph Anders, Raimund W. Kinne.

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
