## [Decision Letter · Decision Letter 0]

11 Mar 2021

PONE-D-21-00965

Temporal and spatial relationship between gluteal muscle Surface EMG activity and the vertical component of the ground reaction force during walking

PLOS ONE

Dear Dr. Anders,

Thank you for submitting your manuscript to PLOS ONE. After careful consideration, we feel that it has merit but does not fully meet PLOS ONE’s publication criteria as it currently stands. Therefore, we invite you to submit a revised version of the manuscript that addresses the points raised during the review process.

Overall, reviewers have some concerns on the methodology (e.g., data analysis) and the lack of explanation on the clinical problem (in the introduction and discussion sections). 

We look forward to receiving your revised manuscript.

Kind regards,

Pei-Chun Kao

Academic Editor

PLOS ONE

Journal Requirements:

Reviewers' comments:

Reviewer's Responses to Questions

**Comments to the Author**

1. Is the manuscript technically sound, and do the data support the conclusions?

Reviewer #1: Partly

Reviewer #2: Yes

Reviewer #3: Yes

Reviewer #4: Partly

Reviewer #5: Yes

2. Has the statistical analysis been performed appropriately and rigorously? 

Reviewer #1: No

Reviewer #2: Yes

Reviewer #3: Yes

Reviewer #4: No

Reviewer #5: Yes

3. Have the authors made all data underlying the findings in their manuscript fully available?

Reviewer #1: Yes

Reviewer #2: Yes

Reviewer #3: Yes

Reviewer #4: No

Reviewer #5: Yes

4. Is the manuscript presented in an intelligible fashion and written in standard English?

Reviewer #1: Yes

Reviewer #2: Yes

Reviewer #3: Yes

Reviewer #4: Yes

Reviewer #5: Yes

5. Review Comments to the Author

Reviewer #1: The authors presented a study relating hip surface electromyography measures to the vertical component of ground reaction forces during walking. As the manuscript currently stands, I do not believe it sufficiently expands upon our current knowledge of the role of the gluteal muscles during gait. Specifically, I do not think that the authors sufficiently set up the introduction and discussion to clearly explain the clinical problem, nor what the novelty of the study is and what it currently adds to the literature. Furthermore, I do not this the statistical analysis, in its current state, appropriately accounts for the time series element of human locomotion, and that the data has been over sampled in the paired t-test analysis and may lead to inappropriate interpretations of the findings. Please see specific comments below, thank you for your time.

Abstract: Need to explain more in depth as to what P2, p4, p5, and p7 are referring to in the manuscript, as it is currently unclear.

Introduction: as it stands, the introduction does not paint a very clear picture as to what the study aims to do, other than re-capping the actions of the gluteal muscles and discussing their function during gait. I think there could be more explanation as to what the clinical question is aiming to address specifically that is currently unknown, and what the value of this information would be to the audience to support your question. Additionally, the study's purpose statement as it stands remains fairly vague and does not get into the nuanced relationships that you were able to assess with the number of emg electrodes and subsequent assessments. This section requires specific attention and more on the manuscript's intended impact.

Line 62: Unclear what part iii "subisdence" means. This recurs throughout the paper.

Line 155: was there a GRF threshold for determining heel contact?

Statistical Analysis: This statistical approach is not the most appropriate means to compare time-series data across the gait cycle. Recent literature advocates utilizing statistical parametric mapping t-test analyses which account for the multiple observations over the time series waveform without data reduction, which is a current limitation of the analysis presented here. This also avoids over-sampling with the separate t-tests. Please see for more explanations and examples:

Pataky TC. One-dimensional statistical parametric mapping in Python. Comput

Methods Biomech Biomed Engin. 2012;15(3):295-301.

Pataky TC. Generalized n-dimensional biomechanical field analysis using

statistical parametric mapping. J Biomech. 2010;43(10):1976-1982.

doi:10.1016/j.jbiomech.2010.03.008.

De Ridder R, Willems T, Vanrenterghem J, Robinson M, Pataky T, Roosen P. Gait

kinematics of subjects with ankle instability using a multisegmented foot model.

Med Sci Sports Exerc. 2013;45(11):2129-2136.

doi:10.1249/MSS.0b013e31829991a2.

The same SPM approach can similarly be done for SPM-ANOVAs.

Discussion: It should be mentioned in the discussion about common innervation of the gluteal muscles that is driving concerted motion, and subsequent physiological coupling as identified in the surface level EMG.

Line 267: Can you clarify which specific fibers you were over for your analysis for a more thorough interpretation of the findings?

Line 276: Need to acknowledge contributing factors of other surrounding musculature throughout the entire lower extremity responsible for controlling loading response during gait.

Line 297: Other than this statement, there was no discussion on the contributions of the TFL, and why this was included as the "glute complex" when arguably gluteus minimus, gluteus medius, and gluteus maximus consitute the gluteal complex. Although the TFL is part of the hip, the TFL is more anterolaterally located and has more responsibility for aiding in hip flexion that the other muscles measured. Please delve more into this concept in the discussion.

Line 329: Need to address the limitations of not measuring the gluteus minimus, as well as acknowledging the older age range of the participants that may have fed into the electromechanical delay elements you identified in your study.

Reviewer #2: The current study explores the spatial and temporal relationship between gluteal muscle function and ground reaction forces during stance phase of walking. Thus, a further clarification on specific muscle firing patterns is introduced, which is very interesting for the general movement science community. However, there are some minor comments need to be addressed before consideration for publication.

Lines 115-116: How the speeds are in good agreement with published data? Please rephrase.

Lines 117-121: By “EMG file” you mean the EMG signal? Although you mention that the time window was manually defined, it seems that there is already a method to detect the heelstrikes, mentioned in line 153 (with pressure sensors). Why did you have different methods for detecting heelstrikes? Please elaborate.

Table 2: This table is most confusing. First, it misses a cross reference in the text. Second, could you please elaborate on the magnitudes calculated for GRFs? For example, the relative magnitudes of GRFy in the normal speed seem very high, and also there isn’t supposed to be any units for relative values. Also, the relative amplitudes for all GRFs have to be 100 for the slow speed (like GRFz), since this is the speed you normalize to, right?

Reviewer #3: 1. Summary of the research

This paper analyzes spatial and temporal relationships between muscle activities in gluteal regions and the vertical component of the GRF during gait cycles. The study was performed on older subjects (mean age about 60 years) with three different individual (slow, normal and fast) walking speeds and surface EMG (SEMG) pads were used. In particular, the occurrence of the respective maximal value concerning the load response (here denoted as F1 peak) is compared to the respective peak of the vertical GRF as only the F1 peak is clearly identifiable throughout all measurements. The results demonstrate the impact on amplitude and time with increasing walking speed and conclude that the F1 peak of the SEMG data proceeds from dorsal to ventral. Also, the highest overall correlation between SEMG data and the vertical GRF can be found in the tensor fasciae latae muscle (TFL).

This study contains some interesting insights on SEMG data and their correlation with a specific set of muscles. The paper is also well structured and overall well written.

This reviewer has some minor comments:

2. Examples and evidence

Minor issues

1. (Abstract) The locations of P1-P8 as well as the F1 peak, denoting the load response, were used before their definition. This reviewer suggests removing/referencing the placement identifier Pi (in the abstract, as it does not contain any relevant information at this point) and replacing the F1 peak by a paraphrase to avoid confusion.

2. Line 69: Please explain the importance of analyzing the GRF and include clinical applications.

3. Line 71: What are the anterior parts of the gluteal muscle unit? Are the deeper located psoas major and iliacus also contained (as this rewiever assumes that these muscles also contribute to the swing phase)?

4. Line 115: Please clarify which type of motion tracking system was used. (Optical marker based tracking, inertial tracking, etc.?)

5. Lines 134 - 406: The authors use “Bonferroni correction” and “Bonferroni procedure” (or “Bonferroni-Holm”, respectively). If it describes the same procedure (in particular, the two outside the brackets), this reviewer suggests being more consistent regarding this.

6. (Discussion) This reviewer suggests adding some possible clinical applications the authors have in mind.

Reviewer #4: I appreciate the opportunity to review this study, which aimed to analyze the temporal relationship between surface EMG of the gluteal region and the corresponding ground reaction force, based on the premise that optimized temporal and spatial activation of the gluteal intermuscular functional unit is essential for steady gait and minimized joint loading. To address this issue, the authors analyzed vertical ground reaction force and surface EMG (gluteal region) in healthy subjects during three different self-paced walking speeds (slow, normal and fast).

The study is relevant and deals with an interesting topic, but there are some aspects that need to be clarified to be suitable for publication. I would like to highlight two major issues that could influence the interpretation and validity of the results: the way that steady walking conditions were identified and the lack of information about data variability and confidence intervals.

Steady walking condition is a central issue of the study. However, it was visually identified by one single person. I don’t understand the reason, since the authors have a motion tracking system, used to obtain walking speed. I suggest using the motion tracking system to calculate the acceleration to identify the steady walking condition or, if this is not possible, to present some measure of error / reproducibility of the visual identification. Since the authors are working with the averaged cycles, it is very important to identify steady walking with accuracy.

The authors affirm that “The current study provides normative data in a healthy older population, which may now be used for future investigations concerning age-related influences, improved functional diagnostics, and the monitoring of therapeutic and rehabilitative efforts.” (page 21, lines 311-313). However, only the average curves are presented, with no information on the variability of these measures. In my view, although the average curve has identified some pattern, knowing the variability is as important as knowing the average value. Standard deviations are shown in figure 2 only for the peaks and are quite high. This makes it difficult to believe in the relevance of the differences shown in the amplitudes between the different speeds, for example. I suggest presenting the standard deviation of the curves as well, the confidence interval and the effect size of the differences found. Just saying that p < 0.05 is not enough to fully understand the results.

Minor issues

1. Page 12 line 113: I did not understand the reason for citation 16 in the results of fast speed

2. Page 12 line 115: please, explain in more detail how the walking speeds were calculated (Mean speed? Or averaged instantaneous speed? Considered the speed of the center of mass or another point?)

3. Page 13: Table 2 should be in the results session

4. Page 14: I suggest including an image of the placement of the electrodes. Readers cannot be forced to use another publication to access this information.

5. Page 14: Clarify whether the force platforms, EMG and pressure sensors were synchronized.

6. Page 17 line 217: “…, the SEMG F1 peak occurred earlier with increasing speed (Fig. 1A)”. This is remarkable for the TFL, but much less evident for Gmed and Gmax, which show a very small difference, raising doubts about its relevance. I suggest that the authors add something about this in the text.

7. The differences indicated by colored bars are nor clear. You can understand in which part of the cycle there was a difference, but you cannot understand who is different from who.

Reviewer #5: Overall Summary

Thank you for the opportunity to review this paper, which investigated the temporal and spatial relationship of SEMG of the gluteal muscles with vGRF in 54 older adults at three different walking speeds.

The authors present an interesting manuscript on an understudied topic. It was good to read, and the general focus of the manuscript is clear. The methods used are appropriate to answer the research question posed, and the limitations of the study are adequately reported and addressed. However, the lack of a clear conclusion in the main body of the manuscript lets down what is otherwise a well-written study. Overall, this seems a good piece of experimental work that has the potential, with some clarifications, to help understand some mechanisms underlying the control of walking gait.

I only have a few specific comments.

Methods

I would recommend the term ‘participants’ as opposed to ‘subjects’ throughout when referring to human research volunteers.

Line 199-200: “Also, the relative changes of the F1 peak after normalization to the slow walking speed were analyzed” – how? Presumably using the same method described in lines 196-199? Please clarify.

Line 203-204: Do you mean to say here that the majority of statistical analysis (i.e. all the results presented in your figures) was performed on the left side only? This took a few reads – please consider rewording.

Discussion/Conclusions

The discussion tails off somewhat with no clear concluding statement. I would suggest adding a conclusion to summarise the key study findings, implications and further work after the limitations section.

Figures

Fig. 2 seems to be missing the legend denoting colour of slow/ normal/ fast walking speed.

6. PLOS authors have the option to publish the peer review history of their article (what does this mean?). If published, this will include your full peer review and any attached files.

Reviewer #1: No

Reviewer #2: **Yes: **Georgios Giarmatzis, PhD

Reviewer #3: No

Reviewer #4: No

Reviewer #5: No

---

## [Author Response · Author response to Decision Letter 0]

6 Apr 2021

PONE-D-21-00965

Temporal and spatial relationship between gluteal muscle Surface EMG activity and the vertical component of the ground reaction force during walking

We would like to thank all reviewers for their time and effort. We have attempted to incorporate all of their recommendations and hope that the manuscript is now suitable for publication in PLOS One. The point to point answers to the reviewers' comments appear below.

Reviewer #1: The authors presented a study relating hip surface electromyography measures to the vertical component of ground reaction forces during walking. As the manuscript currently stands, I do not believe it sufficiently expands upon our current knowledge of the role of the gluteal muscles during gait. Specifically, I do not think that the authors sufficiently set up the introduction and discussion to clearly explain the clinical problem, nor what the novelty of the study is and what it currently adds to the literature. Furthermore, I do not this the statistical analysis, in its current state, appropriately accounts for the time series element of human locomotion, and that the data has been over sampled in the paired t-test analysis and may lead to inappropriate interpretations of the findings. Please see specific comments below, thank you for your time.

Abstract: Need to explain more in depth as to what P2, p4, p5, and p7 are referring to in the manuscript, as it is currently unclear.

Answer

According to your comment and the comment of reviewer #3, we have changed the wording in the abstract by now defining the electrode positions. (lines 35-39)

Introduction: as it stands, the introduction does not paint a very clear picture as to what the study aims to do, other than re-capping the actions of the gluteal muscles and discussing their function during gait. I think there could be more explanation as to what the clinical question is aiming to address specifically that is currently unknown, and what the value of this information would be to the audience to support your question. Additionally, the study's purpose statement as it stands remains fairly vague and does not get into the nuanced relationships that you were able to assess with the number of emg electrodes and subsequent assessments. This section requires specific attention and more on the manuscript's intended impact.

Answer

The specific aim of the study, the incremental advantages of the simultaneous temporal analysis of the spatially resolved SEMG of the gluteal region and the corresponding GRF, and potential clinical implications are now emphasized in more detail in the Introduction. We thank the reviewer for this helpful comment. (lines 100-109)

Line 62: Unclear what part iii "subisdence" means. This recurs throughout the paper.

Answer

For clarity, we have changed the respective sections in the Introduction. (Lines 69-70)

Line 155: was there a GRF threshold for determining heel contact?

Answer

Heel contacts were determined if GRFz values exceeded 20 Newton (as pre-determined in the Vicon 460 system). This is now stated in the Methods (Lines 141-142).

Statistical Analysis: This statistical approach is not the most appropriate means to compare time-series data across the gait cycle. Recent literature advocates utilizing statistical parametric mapping t-test analyses which account for the multiple observations over the time series waveform without data reduction, which is a current limitation of the analysis presented here. This also avoids over-sampling with the separate t-tests. Please see for more explanations and examples:

Pataky TC. One-dimensional statistical parametric mapping in Python. Comput

Methods Biomech Biomed Engin. 2012;15(3):295-301.

Pataky TC. Generalized n-dimensional biomechanical field analysis using

statistical parametric mapping. J Biomech. 2010;43(10):1976-1982.

doi:10.1016/j.jbiomech.2010.03.008.

De Ridder R, Willems T, Vanrenterghem J, Robinson M, Pataky T, Roosen P. Gait

kinematics of subjects with ankle instability using a multisegmented foot model.

Med Sci Sports Exerc. 2013;45(11):2129-2136.

doi:10.1249/MSS.0b013e31829991a2.

Answer

According to the provided literature and other publications, the SPM methodology basically uses single t-tests. The same applies to the Bonferroni-Holm method.

If we understand your comment correctly, your main concern is the accumulation of Type I errors due to multiple testing. We agree that multiple testing increases the probability of Type I errors.

We have carefully compared the results using our statistical approach to those obtained when applying the SPM method. It turns out that the two methods lead to identical results, e.g. concerning both curve characteristics and time range of significant differences between the slow versus normal walking speed at the P2 position (please see the graphs below for comparison). Please note that for the comparison of the two methods we had to transform our p values to quantitatively similar values as in the SPM method by calculating the decimal logarithms of the reciprocals of the p values.

In summary, the two methods applied to detect significant differences between the time series data lead to identical results and are thus equally suitable. Therefore, we thank the reviewer for this important comment, but would prefer not to change the statistical methodology or the graphical representation in the figures.

If required, the continuous display of statistical probability data can be added as a total of 12 graphs (GRF and 3 electrode positions; 3 walking speeds each) in the supplementary data.

 

Exemplary result of the SPM Method (grey areas indicate significant differences) for the comparison of slow vs. normal at the P2 position

Exemplary result of the Bonferroni-Holm Method (p values transformed as described above, reference lines indicate significant differences) for the comparison of slow vs. normal at the P2 position

 

The same SPM approach can similarly be done for SPM-ANOVAs.

Answer

Thank you for this hint. At this point, we interpret you comment more towards future applications and less towards changes of the actual manuscript, since only one ANOVA was performed for the values at the F1 peak of the SEMG data and thus not for time series data.

Discussion: It should be mentioned in the discussion about common innervation of the gluteal muscles that is driving concerted motion, and subsequent physiological coupling as identified in the surface level EMG.

Answer

We agree that the gluteal region is to be considered as a finely tuned functional intermuscular unit. We have substantially expanded the respective parts in the Discussion and the Conclusions. (lines 360-363, 396-408)

Line 267: Can you clarify which specific fibers you were over for your analysis for a more thorough interpretation of the findings?

Answer

We have performed cadaver investigations, already published in a preceding article also published in PlosOne (Anders C, Patenge S, Sander K et al. Detailed spatial characterization of superficial hip muscle activation during walking: A multi-electrode surface EMG investigation of the gluteal region in healthy older adults. Plos One 2017; 12. doi:ARTN e0178957. 10.1371/journal.pone.0178957). Adding this specific information would therefore double already published data.

As already mentioned in the methods section, electrodes were horizontally aligned at half distance between the greater trochanter and the iliac crest, thus covering fibers of tensor fasciae latae (P1,P2), gluteus medius (P3 to P6), and gluteus maximus muscles (P7, P8). We would therefore prefer to reference these findings (see lines 170-177, please see also the new Figure 1).

Line 276: Need to acknowledge contributing factors of other surrounding musculature throughout the entire lower extremity responsible for controlling loading response during gait.

Answer

We have added the respective information in the Introduction (lines 75-76)

Line 297: Other than this statement, there was no discussion on the contributions of the TFL, and why this was included as the "glute complex" when arguably gluteus minimus, gluteus medius, and gluteus maximus consitute the gluteal complex. Although the TFL is part of the hip, the TFL is more anterolaterally located and has more responsibility for aiding in hip flexion that the other muscles measured. Please delve more into this concept in the discussion.

Answer

The role of the TFL and its potential involvement in different phases of the gait cycle are now more extensively addressed in the Introduction and the Discussion (lines75-81, 346-352

Line 329: Need to address the limitations of not measuring the gluteus minimus, as well as acknowledging the older age range of the participants that may have fed into the electromechanical delay elements you identified in your study.

Answer

We have now added the above-mentioned aspects in the limitations section and have also expanded the references accordingly. (lines 372-376)

 

Reviewer #2: The current study explores the spatial and temporal relationship between gluteal muscle function and ground reaction forces during stance phase of walking. Thus, a further clarification on specific muscle firing patterns is introduced, which is very interesting for the general movement science community. However, there are some minor comments need to be addressed before consideration for publication.

Lines 115-116: How the speeds are in good agreement with published data? Please rephrase.

Answer

According to the comment we have rephrased the section for clarity. (lines 139-143)

Lines 117-121: By “EMG file” you mean the EMG signal? Although you mention that the time window was manually defined, it seems that there is already a method to detect the heelstrikes, mentioned in line 153 (with pressure sensors). Why did you have different methods for detecting heelstrikes? Please elaborate.

Answer

To be exact, during the measurement the examiner pressed a keyboard button just after the first two steps and another one just before subjects decelerated at the end of the walkway. These time marks were added to the actual measurement file that contained the SEMG signals, i.e. the SEMG data. 

We have thus changed term "file" to "data" for clarity (line 147).

I addition, your comment about the detection of the heel strikes requires clarification. The embedded force plates provided three-dimensional GRF data that were used to calculate the GRF curves. By using the GRF data, only the time points for those heel strikes occurring on the force plates could be determined (depending on how often the force plates were hit completely). Since these force plates were located in the center of the walkway, maximally one complete stride per side (and thus two heel strikes per side) and per trial could be detected during a total of eight to ten completed trials.

Because of the stochastic nature of the SEMG signal, however, we needed many more strides than only eight to ten for reliable SEMG analysis. Although according to the literature a minimum of twelve complete strides appears sufficient, capturing more strides is always desirable. We therefore placed pressure sensors below the heels of every subject that were used to detect every single heel strike event along the complete walkway. There may be a small, but systematic error between the two detection methods for heel strikes of maximally 1.0 to 0.5 ms for the slow to fast walking speeds. However, since the range of the TD between the respective F1 peaks in SEMG and GRF is in the range of 160 to 70 ms (max. 0.75% error), this does not question the validity of the present results. This is now emphasized in the Limitations section. (lines 377-387)

Table 2: This table is most confusing. First, it misses a cross reference in the text. Second, could you please elaborate on the magnitudes calculated for GRFs? For example, the relative magnitudes of GRFy in the normal speed seem very high, and also there isn’t supposed to be any units for relative values. Also, the relative amplitudes for all GRFs have to be 100 for the slow speed (like GRFz), since this is the speed you normalize to, right?

Answer

Table 2 emphasizes the relative amplitudes of the respective F1 components of all GRF directions together with its temporal occurrence for all investigated walking speeds. It was included to support the statements of the previous paragraph with respect to the selection of the GRFz component for further analysis. The respective reference for the table is provided at the end of the previous paragraph.

All GRF values were calculated as relative values according to the individual body weight (n.b.: in good agreement with published data). This is displayed below the mentioned parameters in square brackets.

Also, the occurrence times of the F1 peaks are expressed as relative times with respect to the whole gait cycle (also mentioned in square brackets). In addition, for the normal and fast walking speeds the differences (relative time points) vs. slow walking speeds are provided to support the statements concerning the time shifts of the F1 component in the preceding parts of the text. To improve the clarity of the reported values, the heading and the formatting of the table has been changed. 

Reviewer #3:

1. Summary of the research

This paper analyzes spatial and temporal relationships between muscle activities in gluteal regions and the vertical component of the GRF during gait cycles. The study was performed on older subjects (mean age about 60 years) with three different individual (slow, normal and fast) walking speeds and surface EMG (SEMG) pads were used. In particular, the occurrence of the respective maximal value concerning the load response (here denoted as F1 peak) is compared to the respective peak of the vertical GRF as only the F1 peak is clearly identifiable throughout all measurements. The results demonstrate the impact on amplitude and time with increasing walking speed and conclude that the F1 peak of the SEMG data proceeds from dorsal to ventral. Also, the highest overall correlation between SEMG data and the vertical GRF can be found in the tensor fasciae latae muscle (TFL).

This study contains some interesting insights on SEMG data and their correlation with a specific set of muscles. The paper is also well structured and overall well written.

This reviewer has some minor comments:

2. Examples and evidence

Minor issues

1. (Abstract) The locations of P1-P8 as well as the F1 peak, denoting the load response, were used before their definition. This reviewer suggests removing/referencing the placement identifier Pi (in the abstract, as it does not contain any relevant information at this point) and replacing the F1 peak by a paraphrase to avoid confusion.

Answer

We have changed the wording in the abstract on the basis of the comment. (lines 35-39)

2. Line 69: Please explain the importance of analyzing the GRF and include clinical applications.

Answer

We have now added a statement about the clinical relevance and application. For better readability, we have added this paragraph in lines 89-91 and at the end of the Introduction (lines 110 - 117).

3. Line 71: What are the anterior parts of the gluteal muscle unit? Are the deeper located psoas major and iliacus also contained (as this rewiever assumes that these muscles also contribute to the swing phase)?

Answer

As recommended, we have added the respective information. (lines 80-81)

4. Line 115: Please clarify which type of motion tracking system was used. (Optical marker based tracking, inertial tracking, etc.?)

Answer

We used an optical marker tracking system. This information has now been added. (lines 139-143)

5. Lines 134 - 406: The authors use “Bonferroni correction” and “Bonferroni procedure” (or “Bonferroni-Holm”, respectively). If it describes the same procedure (in particular, the two outside the brackets), this reviewer suggests being more consistent regarding this.

Answer

We have now changed the wording accordingly: for the Bonferroni correction method we now use "Bonferroni correction" throughout the manuscript. For the Bonferroni-Holm correction we use "Bonferroni-Holm procedure", since it reflects a different stepwise correction procedure, treating every single p-value slightly different with respect to its rank among all considered p-values. Although the details of the Bonferroni-Holm procedure are already referenced in the section “Statistical analyses” of the “Methods”, we can provide a detailed description of the Bonferroni-Holm procedure, if considered relevant. Please also refer to our respective answer to reviewer#1.

6. (Discussion) This reviewer suggests adding some possible clinical applications the authors have in mind.

Answer

We have now added clinical implications as recommended (lines 360-363).

 

Reviewer #4: I appreciate the opportunity to review this study, which aimed to analyze the temporal relationship between surface EMG of the gluteal region and the corresponding ground reaction force, based on the premise that optimized temporal and spatial activation of the gluteal intermuscular functional unit is essential for steady gait and minimized joint loading. To address this issue, the authors analyzed vertical ground reaction force and surface EMG (gluteal region) in healthy subjects during three different self-paced walking speeds (slow, normal and fast).

The study is relevant and deals with an interesting topic, but there are some aspects that need to be clarified to be suitable for publication. I would like to highlight two major issues that could influence the interpretation and validity of the results: the way that steady walking conditions were identified and the lack of information about data variability and confidence intervals.

Steady walking condition is a central issue of the study. However, it was visually identified by one single person. I don’t understand the reason, since the authors have a motion tracking system, used to obtain walking speed. I suggest using the motion tracking system to calculate the acceleration to identify the steady walking condition or, if this is not possible, to present some measure of error / reproducibility of the visual identification. Since the authors are working with the averaged cycles, it is very important to identify steady walking with accuracy.

Answer

The walkway was equipped with a reflective sensor-based motion tracking system that was calibrated for its central part, i.e. the part containing the embedded force plates (Vicon, Oxford). While walking over the force plates, the time intervals between subsequent steps were determined. By additionally using the distance information provided by the optical tracking system, walking speeds were calculated. Therefore, the GRF data and the walking speeds were determined once per trial (at the time of investigation not allowing the derivation of the acceleration for the whole walkway trial).

We had to use the gait analysis and the SEMG systems separately, since at the time of the investigation no plugin was available for the applied SEMG measurement system. This was mainly due to the applied SEMG system, which is far beyond standard SEMG applications for clinical routine, because of the number of channels and the monopolar technique. Therefore, although simultaneously applied, the two measurement systems gathered their data separately. 

Due to the stochastic nature of SEMG signals, we tried to use as many valid strides as possible, and put a high effort in ensuring steady walking conditions. This contained: i) placing start and end markers during each trial; ii) only using strides which did not deviate more than ± 10% from the respective individual median stride time (from all complete strides of every walking speed); and iii) only using strides whose time- normalized SEMG curves did not exceed more than ± 2 SD of all respective strides per subject and walking speed and separately for every single SEMG channel. The set start and stop markers were used to identify analysis regions (time window between start and stop marker). Therefore, the set markers did not define steady strides, but the analysis region. Since ten or more trials per walking speed were performed during the investigation, many more data points were captured than used for analysis. Using this additional event-related information, we were able to only analyse reliable strides (please compare with lines 144-148 and lines 200-204 of the revised manuscript).

The authors affirm that “The current study provides normative data in a healthy older population, which may now be used for future investigations concerning age-related influences, improved functional diagnostics, and the monitoring of therapeutic and rehabilitative efforts.” (page 21, lines 311-313). However, only the average curves are presented, with no information on the variability of these measures. In my view, although the average curve has identified some pattern, knowing the variability is as important as knowing the average value. Standard deviations are shown in figure 2 only for the peaks and are quite high. This makes it difficult to believe in the relevance of the differences shown in the amplitudes between the different speeds, for example. I suggest presenting the standard deviation of the curves as well, the confidence interval and the effect size of the differences found. Just saying that p < 0.05 is not enough to fully understand the results.

Answer

In the original submission, we did not provide SD values for figures 1 and S1 displaying the grand averaged curves for clarity of display. We have now added 95% CI intervals for these two figures. We can also provide the respective effect size values, but this would require additional figures. 

Minor issues

1. Page 12 line 113: I did not understand the reason for citation 16 in the results of fast speed

Answer

This particular reference was inserted to indicate that part of the data was already presented elsewhere. To avoid multiple citations, we provided the reference only once for the fast walking speed.

2. Page 12 line 115: please, explain in more detail how the walking speeds were calculated (Mean speed? Or averaged instantaneous speed? Considered the speed of the center of mass or another point?)

Answer

Your comment revealed a writing error in the methods section. Walking speeds were determined as the subjects walked over the embedded force plates in the middle of the walkway. Heel strike events were determined if the GRFz values exceeded 20 Newton. The optical motion tracking system provided path information of representative markers on the subject's body. By using the time and path information, the walking speed was calculated. This information has now been added to the methods. (lines 139-143)

3. Page 13: Table 2 should be in the results session

Answer

Table 2 only serves to explain, why we selectively analysed the F1 component of the GRFz direction. As this particular aspect is explained and discussed in the methods section mainly as a pre-analysis step, we would prefer to leave it there.

4. Page 14: I suggest including an image of the placement of the electrodes. Readers cannot be forced to use another publication to access this information.

Answer

We have now added a new Figure 1 showing the electrodes and their positions.

5. Page 14: Clarify whether the force platforms, EMG and pressure sensors were synchronized.

Answer

The force platforms were located in the centre of the walkway. Therefore, only one complete stride per trial could be detected. SEMG data were sampled while participants walked along the whole walkway. Both systems worked independently of each other, but the data were sampled simultaneously. This may lead to slight differences between the time points detected for specific heel strike events by the two separate systems. This is now discussed in detail in the limitations section (lines 377-387; please also compare with our answer to the respective comment of reviewer #2).

6. Page 17 line 217: “…, the SEMG F1 peak occurred earlier with increasing speed (Fig. 1A)”. This is remarkable for the TFL, but much less evident for Gmed and Gmax, which show a very small difference, raising doubts about its relevance. I suggest that the authors add something about this in the text.

Answer 

We agree with the reviewer that most pronounced changes occurred for the comparison of the anterior with middle and posterior parts. To address this comment, we have now mentioned Figure 3 earlier in the results section. In addition, this issue is mentioned in the discussion section (Lines 340-345).

7. The differences indicated by colored bars are nor clear. You can understand in which part of the cycle there was a difference, but you cannot understand who is different from who.

Answer

To improve the clarity of the figures 2 and S1, the graphic formatting of the symbols indicating statistically significant differences and the respective legend have been changed.

 

Reviewer #5: Overall Summary

Thank you for the opportunity to review this paper, which investigated the temporal and spatial relationship of SEMG of the gluteal muscles with vGRF in 54 older adults at three different walking speeds.

The authors present an interesting manuscript on an understudied topic. It was good to read, and the general focus of the manuscript is clear. The methods used are appropriate to answer the research question posed, and the limitations of the study are adequately reported and addressed. However, the lack of a clear conclusion in the main body of the manuscript lets down what is otherwise a well-written study. Overall, this seems a good piece of experimental work that has the potential, with some clarifications, to help understand some mechanisms underlying the control of walking gait.

I only have a few specific comments.

Methods

I would recommend the term ‘participants’ as opposed to ‘subjects’ throughout when referring to human research volunteers.

Answer:

We have changed the wording throughout the manuscript as recommended.

Line 199-200: “Also, the relative changes of the F1 peak after normalization to the slow walking speed were analyzed” – how? Presumably using the same method described in lines 196-199? Please clarify.

Answer:

We have now explained the calculation more in detail (lines 236-238).

Line 203-204: Do you mean to say here that the majority of statistical analysis (i.e. all the results presented in your figures) was performed on the left side only? This took a few reads – please consider rewording.

Answer:

We have reworded the respective section for clarity. (lines 241-242)

Discussion/Conclusions

The discussion tails off somewhat with no clear concluding statement. I would suggest adding a conclusion to summarise the key study findings, implications and further work after the limitations section.

Answer: A conclusion section summarizing and emphasizing the main findings has now been introduced. (lines 396-408)

Figures

Fig. 2 seems to be missing the legend denoting colour of slow/ normal/ fast walking speed.

Answer:

Thank you for this hint – we have now added the missing information.

---

## [Decision Letter · Decision Letter 1]

28 Apr 2021

PONE-D-21-00965R1

Temporal and spatial relationship between gluteal muscle Surface EMG activity and the vertical component of the ground reaction force during walking

PLOS ONE

Dear Dr. Anders,

Thank you for submitting your manuscript to PLOS ONE. After careful consideration, we feel that it has merit but does not fully meet PLOS ONE’s publication criteria as it currently stands. Therefore, we invite you to submit a revised version of the manuscript that addresses the points raised during the review process.

This manuscript has been greatly improved. There is one comment from Reviewer 4 that requires the authors to address. 

We look forward to receiving your revised manuscript.

Kind regards,

Pei-Chun Kao

Academic Editor

PLOS ONE

Journal Requirements:

Additional Editor Comments (if provided):

The manuscript has been greatly improved. There is one more comment from Reviewer 4 that requires the authors to address.

Reviewers' comments:

Reviewer's Responses to Questions

**Comments to the Author**

1. If the authors have adequately addressed your comments raised in a previous round of review and you feel that this manuscript is now acceptable for publication, you may indicate that here to bypass the “Comments to the Author” section, enter your conflict of interest statement in the “Confidential to Editor” section, and submit your "Accept" recommendation.

Reviewer #1: All comments have been addressed

Reviewer #4: (No Response)

2. Is the manuscript technically sound, and do the data support the conclusions?

Reviewer #1: (No Response)

Reviewer #4: Partly

3. Has the statistical analysis been performed appropriately and rigorously? 

Reviewer #1: (No Response)

Reviewer #4: Yes

4. Have the authors made all data underlying the findings in their manuscript fully available?

Reviewer #1: (No Response)

Reviewer #4: No

5. Is the manuscript presented in an intelligible fashion and written in standard English?

Reviewer #1: (No Response)

Reviewer #4: Yes

6. Review Comments to the Author

Reviewer #1: I would like to thank the authors for addressing all previous comments from myself and from the other reviewers. The manuscript now has a more sound rationale as to the contribution to the literature, and I appreciate the authors addressing the previous concern with the statistical parametric mapping suggestion, with the Bonferroni correction to account for Type 1 errors.

Reviewer #4: The paper has improved, but there are still some points to clarify. The aim of the study was to analyze the temporal relationship between spatially resolved surface EMG of the gluteal region and the corresponding ground reaction force. However, according to the response of the authors to my question about the synchronization of the methods, both systems (force plates and EMG system) worked independently of each other. They affirm in the limitations that there is a systematic detection error between the two detection methods of heel contact - maximally 1.0 to 0.5 ms for the slow to fast walking speeds - but data are not shown. How was this systematic error assessed? If the event correspondent to heel contact in the EMG signal was determined by the pressure sensor, were these systems (EMG and pressure sensors) synchronized?

I am still not convinced about the accuracy of the method used to identify the steady walking condition since, although the authors have explained how they controlled the selection of the strides for analysis, no information about the errors of this method has been provided. This should be mentioned as a limitation of the study.

7. PLOS authors have the option to publish the peer review history of their article (what does this mean?). If published, this will include your full peer review and any attached files.

Reviewer #1: No

Reviewer #4: No

---

## [Author Response · Author response to Decision Letter 1]

29 Apr 2021

Reviewer #1: I would like to thank the authors for addressing all previous comments from myself and from the other reviewers. The manuscript now has a more sound rationale as to the contribution to the literature, and I appreciate the authors addressing the previous concern with the statistical parametric mapping suggestion, with the Bonferroni correction to account for Type 1 errors.

Answer

We would like to thank you for this positive evaluation of the manuscript and the made revisions. 

Reviewer #4: The paper has improved, but there are still some points to clarify. The aim of the study was to analyze the temporal relationship between spatially resolved surface EMG of the gluteal region and the corresponding ground reaction force. However, according to the response of the authors to my question about the synchronization of the methods, both systems (force plates and EMG system) worked independently of each other. They affirm in the limitations that there is a systematic detection error between the two detection methods of heel contact - maximally 1.0 to 0.5 ms for the slow to fast walking speeds - but data are not shown. How was this systematic error assessed? 

Answer

Originally, force plate data were measured at a sampling rate of 1,000/s (one sample = 1ms). Heel strike events for the force plates were detected as their load exceeded 200 N in a vertical direction – this is an embedded function in the gait analysis software and is universally applied. Therefore, this detection algorithm can be considered as a gold standard for heel strike detection using force plates. 

As already stated, heel strike events for the SEMG data were determined independently from the force plates by simultaneously measuring SEMG and force sensor data. These data were captured with a sampling rate of 2,000/s (one sample = 0.5ms). Heel strike events were determined using a custom-made script in MATLAB. This algorithm first determined one third of the 20 largest slope values for every complete walking trial as the detection threshold for this particular trial. If two subsequent time points were at least 500 ms apart from each other, the respective event was used as the heel contact time. We have carefully controlled the MATLAB algorithm and possible detection variances from baseline occurred for a maximum of ± 1 sample, i.e. ± 0.5 ms. Mean stride times varied between 1.3 (slow) and 0.92 s (fast). Due to normalization of the resolution to 1% in both systems, this 0.5 ms inaccuracy equals 0.055% of the stride time. With respect to the TD, that was in the time range of 160 to 70 ms one millisecond absolute time difference would therefore maximally equal 0.7% error. We checked the individual TD values and included the mentioned 0.75% detection error to cover all individual TD values.

We cannot provide any data for absolute or relative delays for the force plate data, since these are not available. 

We therefore think that the description of the error in the present form of the manuscript is sufficient.

If the event correspondent to heel contact in the EMG signal was determined by the pressure sensor, were these systems (EMG and pressure sensors) synchronized?

Answer

Both sensors were measured with the same system, i.e. they were synchronized. We have stated this already in the manuscript (L188-189), but have now changed the wording slightly to be more precise.

I am still not convinced about the accuracy of the method used to identify the steady walking condition since, although the authors have explained how they controlled the selection of the strides for analysis, no information about the errors of this method has been provided. This should be mentioned as a limitation of the study.

Answer

We have now added an additional paragraph about how we ensured the accuracy of steady walking conditions in the limitations section (lines 366-373). 

Since all corrections to ensure steady walking conditions were performed online during data collection and pre-analysis the deviation between the final results with and without the application of these corrections cannot be determined (and thus the error of this method cannot be provided).

---

## [Editor Report · Decision Letter 2]

3 May 2021

Temporal and spatial relationship between gluteal muscle Surface EMG activity and the vertical component of the ground reaction force during walking

PONE-D-21-00965R2

Dear Dr. Anders,

We’re pleased to inform you that your manuscript has been judged scientifically suitable for publication and will be formally accepted for publication once it meets all outstanding technical requirements.

Kind regards,

Pei-Chun Kao

Academic Editor

PLOS ONE

---

## [Editor Report · Acceptance letter]

5 May 2021

PONE-D-21-00965R2 

Temporal and spatial relationship between gluteal muscle Surface EMG activity and the vertical component of the ground reaction force during walking 

Dear Dr. Anders:

I'm pleased to inform you that your manuscript has been deemed suitable for publication in PLOS ONE. Congratulations! Your manuscript is now with our production department. 

Kind regards, 

on behalf of

Dr. Pei-Chun Kao 

Academic Editor

PLOS ONE